# Sexual orientation-related disparities in employment, health insurance, healthcare access and health-related quality of life: a cohort study of US male and female adolescents and young adults

Brittany M Charlton,[1,2,3,4] Allegra R Gordon,[1,2] Sari L Reisner,[2,4,5,6] Vishnudas Sarda,[1] Mihail Samnaliev,[1,2] S Bryn Austin[1,2,3,7]

An abstract of this work was presented at the Society for Adolescent Health and Medicine Annual Meeting and GLMA Annual Conference on LGBT Health.

For numbered affiliations see end of article.

**Correspondence to**
Dr Brittany M Charlton; bcharlton@mail.harvard.edu

## ABSTRACT

**Objective** To investigate sexual orientation-related disparities in employment and healthcare, including potential contributions to health-related quality of life (HRQL).

**Setting** Growing Up Today Study, a USA-based longitudinal cohort that began in 1996; predominantly composed of participants who are white and of middle-to-high socioeconomic positions.

**Participants** 9914 participants 18–32 years old at the most recent follow-up questionnaire.

**Primary outcome measure** In 2013, participants reported if, in the last year, they had been unemployed, uninsured or lacked healthcare access (routine physical exam). Participants completed the EQ-5D-5L, a validated, preference-weighted measurement of HRQL. After adjusting for potential confounders, we used sex-stratified, log-binomial models to calculate the association of sexual orientation with employment, health insurance and healthcare access, while examining if these variables attenuated the sexual orientation-related HRQL disparities.

**Results** Sexual minority women and men were about twice as likely as their respective heterosexual counterparts to have been unemployed and uninsured. For example, the risk ratio (95% CI) of uninsured bisexual women was 3.76 (2.42 to 5.85) and of unemployed mostly heterosexual men was 1.82 (1.30 to 2.54). Routine physical examination was not different across sexual orientation groups (p>0.05). All sexual minority subgroups had worse HRQL than heterosexuals (p<0.05) across the five EQ-5D-5L dimensions (mobility, self-care, usual activities, pain/discomfort and anxiety/depression). Controlling for employment and health insurance did not substantially attenuate the existing sexual orientation-related HRQL disparities.

**Conclusions** Research on sexual orientation-related disparities in employment and healthcare has often been limited to comparisons between cohabiting different-sex and same-sex adult couples, overlooking sexual minority subgroups (eg, bisexuals vs lesbians), non-cohabitating populations and young people. Less is known about sexual

### Strengths and limitations of this study

- ► The cohort has predominantly high social status, so while the findings can examine sexual orientation-related differences, they may underestimate the prevalence of unemployment, lack of health insurance, lack of healthcare access and poor health-related quality of life (HRQL) compared with other populations (eg, low social status, elderly).
- ► There may be other factors that mediate the sexual orientation-related HRQL disparities, including bullying victimisation, social status and others.
- ► Data were cross-sectional and limited on some of our variables, such as health insurance and healthcare access.
- ► The study includes a large sample drawn from young adults living across the USA.
- ► This is the first study to examine the disparities across HRQL domains and to evaluate the role of additional factors including employment, health insurance and healthcare access.

orientation-related disparities in HRQL including potential contributions from employment and healthcare. The current study documents that disparities in employment, health insurance and various HRQL dimensions are pervasive across sexual minority subgroups, non-cohabitating couples and youth in families of middle-to-high socioeconomic positions.

## INTRODUCTION

Nearly half of all sexual minorities (eg, lesbian, gay and bisexual individuals) report employment discrimination in their lifetime.[1] Discrimination, along with other social and economic barriers, can lead to unemployment, and subsequently lack of health insurance and healthcare access. All of these

factors may contribute to poor health-related quality of life (HRQL), which is a critical measure of health status. Comparing HRQL across different subpopulations can highlight disparities, as well as help to evaluate the cost-effectiveness of policies or programmes that reduce such disparities.[2]

Previous research has documented that, compared with heterosexuals, sexual minorities are more likely to be unemployed, lack health insurance and lack healthcare access.[3–13] However, these data are often limited to comparisons between cohabitating different-sex couples and same-sex adult couples, which precludes examination of differences in any other aspects such as among sexual minority subgroups (eg, bisexuals vs lesbians), non-cohabitating groups or young people. A few previous studies have also revealed that sexual minorities, on average, have worse quality of life than heterosexuals.[14–19] But most of these data depend on limited measures of quality of life, including measures such as the Medical Outcomes Study 36-Item Short-Form Health Survey, which does not explicitly assess HRQL, including its multiple dimensions (eg, pain/discomfort, anxiety/depression). Additionally, few of these studies have examined potential mediators of the sexual orientation-related disparities in HRQL.

Documenting such disparities can provide policymakers with evidence to inform legislation that can lessen health inequities. For example, over half of the states across the USA currently have no employment non-discrimination law covering sexual orientation.[20] Using research to document the downstream consequences of unemployment—health insurance, healthcare access and HRQL—can aid policymakers in crafting the necessary legal changes to lessen these inequities, such as federal employment non-discrimination laws. Therefore, the goal of this study was to leverage data from a USA-based longitudinal cohort of adolescents and young adults (ages 18–32) to examine sexual orientation-related disparities in employment, health insurance and healthcare access, while evaluating their contributions to HRQL disparities.

## METHODS
### Study population
For the last 20 years, questionnaire data have been collected annually in the Growing Up Today Study (GUTS) from the female and male offspring of the Nurses' Health Study 2 (NHS2) participants. The first wave of GUTS participants (GUTS1) was enrolled in 1996, when they were 9–14 years of age, and another wave aged 9–16 years old was enrolled in 2004 (GUTS2), making the entire cohort 18–32 years of age at the last questionnaire wave in 2013. We originally sent questionnaires to GUTS1 and GUTS2 in alternating years, but since 2013 we have combined GUTS1 and GUTS2 into a single annual questionnaire. Participants' race/ethnicity is primarily white, and most of their families report a middle-to-high household income (64% of participants had an annual household income during their childhood of ≥\$75 000).

The current analysis was limited to GUTS participants who reported their sexual orientation and information on measures of unemployment, health insurance, healthcare access and HRQL between baseline and at the end of follow-up in 2013 (n=9914).

### Measures
#### Sexual orientation
Detailed information about sexual orientation has been repeatedly collected in GUTS1 and GUTS2 using an item adapted from the Minnesota Adolescent Health Survey,[21] which asks about feelings of attraction and identity. The item reads 'Which of the following best describes your feelings?' with the following response options: completely heterosexual (attracted to persons of the opposite sex), mostly heterosexual, bisexual (equally attracted to men and women), mostly homosexual, completely homosexual (gay/lesbian, attracted to persons of the same sex) and not sure.

Sexual orientation groups were modelled using the 2013 questionnaire data (the same questionnaire year as the latest outcome) as follows: completely heterosexual (reference group), mostly heterosexual, bisexual and lesbian/gay (made up from the mostly homosexual and completely homosexual groups). Missing data were limited and imputed from previous questionnaire waves. Respondents endorsing 'not sure' were excluded (n=69).

#### Unemployment
In 2010 (GUTS1 only) and 2013, participants reported their employment status. Response options included working full time, working part-time, student, volunteering, military, unemployed/laid off/looking for work, staying at home with children/taking care of family, on maternity or family leave, and not working due to illness or disability. We categorised participants as being unemployed/not working due to illness or disability in 2013 if they provided an affirmative response to the 'unemployed/laid off/looking for work' item or the 'not working due to illness or disability' item.

#### Lacking health insurance
The 2013 questionnaire asked participants whether they were covered by any kind of health insurance or healthcare plan (yes; no). We categorised participants as lacking health insurance in 2013 if they reported no such coverage.

#### Lacking healthcare access through routine physical exam
The use of routine physical exams was measured by asking about the timing of the last routine physical exam in 2013. We categorised participants who reported their last routine physical exam occurred >12 months before the questionnaire completion as not having healthcare access through a routine physical exam.

#### Health-related quality of life
The 2013 questionnaire assessed HRQL using the EQ-5D-5L,[22] which is a validated, preference-weighted

measure. EQ-5D-5L is a standardised, generic instrument that is applicable to a wide range of health conditions and appropriate for use with adolescents and young adults. This measure can also be used to calculate quality-adjusted life years for economic analyses with USA-based population weights.

EQ-5D-5L assesses HRQL in five dimensions (mobility, self-care, usual activities, pain/discomfort and anxiety/depression) by having participants report a dimension-specific score (1=no problems, 2=slight problems, 3=moderate problems, 4=severe problems and 5=extreme problems). Based on the EuroQol Group recommendations,[23] each physical functioning dimension was dichotomised into a score of 1 being 'no problems' vs scores of 2–5 being 'any problems'. The anxiety/depression dimension was dichotomised into scores of 1–2 being 'none or slight problems' vs scores of 3–5 being 'moderate, severe, or extreme problems', as has been done previously.[24]

All five dimensions were then used to create a summary HRQL index score.[25] Because EQ-5D-5L value sets are not yet available for the USA, in order to preference-weight the index score for US populations we relied on another value set to map the EQ-5D-5L responses to the previous version, the EQ-5D-3L.[26] This summary results in an index score that is calibrated to reflect the degree to which different health statuses are valued in the US population overall. The index scores for the US population range from the most severe impairment on all five dimensions, termed 'worse than death' (value=−0.109), to full health (value=1.0).[27] Previous research suggests that an index score difference as small as 0.02 points can have a clinically meaningful difference.[28] We also analysed the HRQL index score after dichotomising it as a score of 1 being 'full health' vs <1 being 'not full health'.

## Confounders

Potential confounders included baseline age in years, race/ethnicity (white, another race/ethnicity), childhood socioeconomic position (annual household income from the NHS2 report in 2001 (<$50 000, $50 000–$74 999, $75 000–$99 999, ≥$100 000)), sex/gender, marital status in 2013 (married, not married), region of residence (West, Midwest, South or Northeast) and cohort (GUTS1, GUTS2). If a participant's data were missing for potential confounders, data were imputed from previous questionnaire years; if no such data were available for a participant, then multiple imputation procedures were used.

## Statistical analysis

We first examined cross-sectional mean differences in employment, health insurance, healthcare access and HRQL measures across sexual orientation groups. Multivariate regression from log-binomial models was used for dichotomous outcomes to calculate risk ratios (RR) and 95% CIs. Linear regression with the robust sandwich estimator was used for continuous outcomes to calculate the betas (β) and SEs. In order to account for sibling clusters, we estimated the variance using generalised estimating equations with a compound symmetry working correlation matrix.

We calculated the RRs of experiencing unemployment, lacking health insurance or lacking healthcare access by sexual orientation groups (referent=completely heterosexual), adjusted for potential confounders. Analyses for HRQL measures followed the previously used two-step approach,[29 30] by first dichotomising the index score (1 vs <1) and then using the continuous health index score in analyses restricted to those with lower HRQL (defined as health index scores <1). HRQL models were first adjusted for potential confounders, and then adjusted for employment, health insurance and healthcare access to explore attenuation. Previous research with this cohort[14] suggests possible effect modification of the sexual orientation and HRQL association by sex/gender but not by cohort, so all analyses were stratified by sex/gender and adjusted for cohort. Analyses were conducted using SAS V.9.3.

## Patient and public involvement

The public, including patients and study participants, was not involved in setting the research question or the outcome measures, nor was the public involved in developing plans for the study design, recruitment or implementation. The GUTS research results are regularly reported to study participants, including through newsletters and other communications.

## RESULTS

Of the 9914 participants in our sample, 7.5% were unemployed or not working due to illness or disability, 4.9% were uninsured, and 38.2% lacked healthcare access through a routine physical exam in the last year. As shown in table 1, all of these outcomes varied by sexual orientation identity, with sexual minorities having more unemployment, less health insurance and less healthcare access (all p values <0.01, except among men, where healthcare access was not statistically different across sexual orientation groups). The mean HRQL index score was 0.90 for women and 0.92 for men and varied by sexual orientation identity, with sexual minorities having lower mean HRQL index scores compared with completely heterosexuals among both women and men (p<0.001). Within each of the five HRQL domains, sexual minority women and men were more likely than heterosexuals to report worse health (p values ranged from <0.001 to 0.02, except among men, where self-care was not statistically different across sexual orientation groups).

Table 2 presents the RRs and 95% CIs for the association between sexual orientation and the risk of being unemployed, lacking health insurance or lacking healthcare access, adjusted for potential confounders. Compared with heterosexuals, sexual minority women and men were about twice as likely as their respective heterosexual counterparts to have been unemployed and uninsured. For example, bisexual women were at a higher risk than

**Table 1** Age-standardised characteristics of a cohort of US women and men by sex/gender and sexual orientation (n=9914)

| Female (n=6663) | Completely heterosexual (n=5353) | Mostly heterosexual (n=1037) | Bisexual (n=159) | Lesbian/Gay (n=114) | P values* |
|---|---|---|---|---|---|
| Age at baseline†, mean years (SD), range: 18–32 | 26.1 (3.6) | 26.6 (3.4) | 25.5 (3.6) | 26.4 (3.5) | <0.001 |
| Unemployed/not working due to illness or disability in the last year‡, % (n) | 5.9 (317) | 10.5 (109) | 14.5 (23) | 10.5 (12) | <0.001 |
| Uninsured (lacked health insurance) in the last year‡, % (n) | 3.9 (208) | 5.7 (59) | 13.8 (22) | 4.4 (5) | <0.001 |
| Lacked a routine physical exam in the last year‡, % (n) | 32.0 (1712) | 36.1 (374) | 40.3 (64) | 37.7 (43) | 0.01 |
| HRQL§ dimensions, % (n) | | | | | |
| Mobility ≥slight problems | 3.2 (172) | 5.6 (58) | 13.2 (21) | 10.5 (12) | <0.001 |
| Self-care ≥slight problems | 0.5 (28) | 1.3 (13) | 4.4 (7) | 1.8 (2) | <0.001 |
| Usual activities ≥slight problems | 5.0 (267) | 12.3 (127) | 20.1 (32) | 16.7 (19) | <0.001 |
| Pain/discomfort ≥slight problems | 26.5 (1418) | 36.8 (382) | 42.1 (67) | 44.7 (51) | <0.001 |
| Anxiety/depression ≥moderate problems | 12.3 (658) | 22.6 (234) | 36.5 (58) | 29.0 (33) | <0.001 |
| HRQL index score¶, mean (SD) | 0.91 (0.09) | 0.87 (0.09) | 0.84 (0.11) | 0.85 (0.10) | <0.001 |
| Less than full health (HRQL index score <1) | 54.6 (2924) | 73.3 (760) | 84.3 (134) | 79.0 (90) | <0.001 |
| HRQL index score among those with less than full health, mean (SD) | 0.84 (0.06) | 0.83 (0.06) | 0.80 (0.09) | 0.81 (0.08) | <0.001 |
| **Male (n=3251)** | **(n=2805)** | **(n=268)** | **(n=25)** | **(n=153)** | |
| Age at baseline†, mean years (SD), range: 18–32 | 25.9 (3.7) | 26.1 (3.7) | 24.6 (3.8) | 25.9 (3.6) | 0.26 |
| Unemployed/not working due to illness or disability in the last year‡, % (n) | 7.8 (218) | 15.3 (41) | 8.0 (2) | 11.1 (17) | <0.001 |
| Uninsured (lacked health insurance) in the last year‡, % (n) | 5.2 (147) | 9.7 (26) | 0.0 (0) | 11.1 (17) | <0.001 |
| Lacked a routine physical exam in the last year‡, % (n) | 48.8 (1369) | 47.8 (128) | 36.0 (9) | 54.9 (84) | 0.26 |
| HRQL§ dimensions, % (n) | | | | | |
| Mobility ≥slight problems | 3.1 (86) | 6.3 (17) | 8.0 (2) | 2.6 (4) | 0.02 |
| Self-care ≥slight problems | 0.6 (18) | 1.1 (3) | 4.0 (1) | 0.0 (0) | 0.11 |
| Usual activities ≥slight problems | 3.8 (107) | 10.5 (28) | 8.0 (2) | 5.9 (9) | <0.001 |
| Pain/discomfort ≥slight problems | 25.4 (711) | 38.8 (104) | 32.0 (8) | 24.2 (37) | <0.001 |
| Anxiety/depression ≥moderate problems | 10.2 (285) | 25.8 (69) | 24.0 (6) | 28.1 (43) | <0.001 |
| HRQL index score§, mean (SD) | 0.92 (0.09) | 0.87 (0.11) | 0.86 (0.12) | 0.89 (0.09) | <0.001 |
| Less than full health (HRQL index score <1) | 47.3 (1329) | 70.2 (188) | 76.0 (19) | 68.0 (104) | <0.001 |
| HRQL index score among those with less than full health, mean (SD) | 0.84 (0.06) | 0.82 (0.08) | 0.82 (0.11) | 0.84 (0.06) | <0.001 |

*P values were calculated using analysis of variance for continuous variables and $\chi^2$ test for categorical variables (including those with a zero frequency cell such as the self-care HRQL dimension).
†Multiple imputation used in subsequent analyses for any missing covariates data. Per cent missing: race/ethnicity (1.2%), marital status (0.2%), socioeconomic position (17.8%) and geographical region (0.1%).
‡As reported in 2013.
§Health-related quality of life (HRQL) measured by the EQ-5D-5L, a validated preference-weighted measure for US populations.
¶Possible scores ranged from −0.109 ('worse than death') to 1 ('full health').

heterosexual women of being uninsured (RR 3.76 (95% CI 2.42 to 5.85)), and mostly heterosexual men were more likely than heterosexual men to have been unemployed (RR 1.82 (95% CI 1.30 to 2.54)). The use of a routine physical exam was not statistically different across sexual orientation groups.

Table 3 presents the RRs and 95% CIs for the association between sexual orientation identity and the

**Table 2** Multivariable* risk ratios of experiencing unemployment or lacking health insurance or lacking healthcare access in a cohort of US men and women by sex/gender and sexual orientation (n=9914)

| | Relative risk (95% CI) | | | |
| | Completely heterosexual | Mostly heterosexual | Bisexual | Lesbian/Gay |
|---|---|---|---|---|
| **Female (n=6663)** | (n=5353) | (n=1037) | (n=159) | (n=114) |
| Unemployed/not working due to illness or disability in the last year† | 1.00 (ref) | 1.68 (1.35 to 2.09) | 2.39 (1.56 to 3.65) | 1.84 (1.03 to 3.27) |
| Uninsured (lacked health insurance) in the last year† | 1.00 (ref) | 1.39 (1.04 to 1.86) | 3.76 (2.42 to 5.85) | 1.18 (0.49 to 2.88) |
| Lacked a routine physical exam in the last year† | 1.00 (ref) | 1.12 (1.00 to 1.25) | 1.26 (0.98 to 1.62) | 1.17 (0.86 to 1.58) |
| **Male (n=3251)** | (n=2805) | (n=268) | (n=25) | (n=153) |
| Unemployed/not working due to illness or disability in the last year† | 1.00 (ref) | 1.82 (1.30 to 2.54) | 1.00 (0.25 to 4.06) | 1.49 (0.91 to 2.45) |
| Uninsured (lacked health insurance) in the last year† | 1.00 (ref) | 1.67 (1.10 to 2.54) | NA‡ | 2.21 (1.33 to 3.65) |
| Lacked a routine physical exam in the last year† | 1.00 (ref) | 0.97 (0.81 to 1.16) | 0.77 (0.40 to 1.49) | 1.13 (0.90 to 1.40) |

*Adjusted for age, race/ethnicity, childhood household income (reported by the mother in 2001), geographical region and cohort; multiple imputation used for missing covariates.
†As reported in 2013.
‡Every bisexual man reported health insurance coverage in 2013 so no risk ratio was computed.
NA, not available; ref, reference.

risk of having poorer HRQL (index score <1 vs 1). Model 0 demonstrates that, after accounting for potential confounders, sexual minority women and men had elevated risk of less than full health relative to completely heterosexuals. The addition of unemployment (model 1), lacking health insurance (model 2), lacking routine physical exam (model 3) or all three of these combined (model 4) did not substantially

**Table 3** Multivariable* risk ratios of experiencing less than full health (HRQL index score <1 vs HRQL index score=1) in a cohort of US men and women by sex/gender and sexual orientation (n=9914)

| | Relative risk (95% CI) | | | |
| | Completely heterosexual | Mostly heterosexual | Bisexual | Lesbian/Gay |
|---|---|---|---|---|
| **Female (n=6663)** | (n=5353) | (n=1037) | (n=159) | (n=114) |
| Model 0: sociodemographics | 1.00 (ref) | 1.33 (1.27 to 1.39) | 1.53 (1.42 to 1.65) | 1.42 (1.29 to 1.57) |
| Model 1: model 0 + unemployed | 1.00 (ref) | 1.32 (1.26 to 1.38) | 1.51 (1.40 to 1.62) | 1.41 (1.28 to 1.56) |
| Model 2: model 0 + uninsured | 1.00 (ref) | 1.32 (1.27 to 1.38) | 1.50 (1.39 to 1.61) | 1.42 (1.29 to 1.57) |
| Model 3: model 0 + lacked routine physical exam | 1.00 (ref) | 1.33 (1.27 to 1.39) | 1.53 (1.42 to 1.65) | 1.42 (1.29 to 1.57) |
| Model 4: model 0 + unemployed + uninsured + lacked routine physical exam | 1.00 (ref) | 1.32 (1.26 to 1.38) | 1.48 (1.37 to 1.60) | 1.41 (1.28 to 1.56) |
| **Male (n=3251)** | (n=2805) | (n=268) | (n=25) | (n=153) |
| Model 0: sociodemographics | 1.00 (ref) | 1.46 (1.34 to 1.60) | 1.62 (1.29 to 2.03) | 1.43 (1.27 to 1.60) |
| Model 1: model 0 + unemployed | 1.00 (ref) | 1.43 (1.31 to 1.57) | 1.62 (1.28 to 2.05) | 1.41 (1.26 to 1.58) |
| Model 2: model 0 + uninsured | 1.00 (ref) | 1.46 (1.33 to 1.59) | 1.63 (1.29 to 2.04) | 1.42 (1.26 to 1.59) |
| Model 3: model 0 + lacked routine physical exam | 1.00 (ref) | 1.46 (1.34 to 1.60) | 1.62 (1.29 to 2.03) | 1.43 (1.27 to 1.60) |
| Model 4: model 0 + unemployed + uninsured + lacked routine physical exam | 1.00 (ref) | 1.43 (1.31 to 1.56) | 1.62 (1.28 to 2.05) | 1.41 (1.26 to 1.58) |

*Adjusted for age, race/ethnicity, childhood household income (reported by the mother in 2001), geographical region and cohort; multiple imputation used for missing covariates.
HRQL, health-related quality of life; ref, reference.

**Table 4** Multivariable* linear associations between sexual orientation and HRQL index score in young adulthood among those who reported less than full health (score <1) in a cohort of US men and women by sex/gender (n=5547)

| | β (SE) | | | | | |
| | Mostly heterosexual | | Bisexual | | Lesbian/Gay | |
| Female (n=3908) | (n=760) | P values | (n=134) | P values | (n=90) | P values |
|---|---|---|---|---|---|---|
| Model 0: sociodemographics | –0.010 (0.003) | <0.001 | –0.034 (0.008) | <0.001 | –0.026 (0.008) | 0.002 |
| Model 1: model 0 + unemployed | –0.010 (0.003) | <0.001 | –0.033 (0.008) | <0.001 | –0.025 (0.008) | 0.002 |
| Model 2: model 0 + uninsured | –0.010 (0.003) | <0.001 | –0.033 (0.008) | <0.001 | –0.026 (0.008) | 0.002 |
| Model 3: model 0 + lacked routine physical exam | –0.010 (0.003) | <0.001 | –0.034 (0.008) | <0.001 | –0.025 (0.008) | 0.002 |
| Model 4: model 0 + unemployed + uninsured + lacked routine physical exam | –0.009 (0.003) | <0.001 | –0.033 (0.008) | <0.001 | –0.025 (0.008) | 0.002 |
| Male (n=1639) | (n=188) | | (n=19) | | (n=104) | |
| Model 0: sociodemographics | –0.020 (0.006) | 0.001 | –0.021 (0.025) | 0.40 | –0.004 (0.006) | 0.47 |
| Model 1: model 0 + unemployed | –0.019 (0.006) | 0.002 | –0.022 (0.025) | 0.38 | –0.003 (0.006) | 0.54 |
| Model 2: model 0 + uninsured | –0.020 (0.006) | 0.001 | –0.022 (0.025) | 0.38 | –0.004 (0.006) | 0.54 |
| Model 3: model 0 + lacked routine physical exam | –0.020 (0.006) | 0.001 | –0.021 (0.025) | 0.40 | –0.004 (0.006) | 0.47 |
| Model 4: model 0 + unemployed + uninsured + lacked routine physical exam | –0.019 (0.006) | 0.002 | –0.022 (0.025) | 0.38 | –0.003 (0.006) | 0.58 |

*Adjusted for age, race/ethnicity, childhood household income (reported by the mother in 2001), geographical region and cohort; multiple imputation used for any missing covariates; completely heterosexual is the reference.

attenuate the associations between sexual orientation and HRQL.

Table 4 presents the results for the multivariable linear regression of HRQL restricted to those with less than full health (index score <1). The patterns were similar to those provided in table 3, with sexual minorities having lower HRQL compared with their completely heterosexual peers, and these relationships were not substantially attenuated after accounting for the effects of unemployment, lacking health insurance and lacking a routine physical exam.

## DISCUSSION

Sexual minority women and men are more likely than heterosexuals to have been unemployed and uninsured in early adulthood. Within each of the five HRQL dimensions, sexual minorities were also more likely than their heterosexual peers to report worse health. These disparities are pervasive in a US population that predominantly holds high social status with regard to race/ethnicity and socioeconomic position.

The existing literature on sexual orientation-related employment disparities primarily focuses on its contribution to more poverty among sexual minorities compared with heterosexuals.[4–6 13] Sexual minority women, as well as sexual minority people of colour, appear to be especially vulnerable to poverty. Badgett and Schneebaum[5] documented that policies to reduce wage gaps between heterosexual men and various sexual minority groups, including women and people of colour, can significantly reduce poverty. Although there is limited research on sexual orientation and employment status, data from the 2010 American Community Survey compare male same-sex couples and female same-sex couples with different-sex couples, all of which are cohabiting. These data reveal that cohabiting women in same-sex couples are more likely than women or men in cohabitating different-sex couples to be unemployed or not in the labour force. An estimated 40.4% of cohabiting women in same-sex couples were unemployed or not in the workforce compared with 21.8% of cohabitating men in same-sex relationships and 25.5% of cohabitating women and men in different-sex couples. Women in same-sex couples benefit the least from employment as a way out of poverty because even among employed groups, women are at the highest risk of being in poverty.[6] The

findings of the present study support these data, revealing higher unemployment among sexual minority women compared with heterosexual women. We also found larger employment disparities among sexual minority men, possibly because our sample was not restricted to cohabitating participants.

Other studies have documented sexual orientation-related disparities in health insurance and healthcare access. For example, one of the largest nationally representative samples recently revealed that women in same-sex relationships were significantly less likely than women in different-sex relationships to have health insurance or to have had a check-up in the last year. Men in same-sex relationships in that sample were also less likely than men in different-sex relationships to have health insurance but more likely to have a check-up in the last year.[7] The present study supports most of the same patterns and a similar magnitude of health insurance and healthcare access disparities. A number of other studies document these same health insurance and healthcare access disparities using a cohabitating-based approach,[3 9 11 12] and these results have also been repeated in other non-cohabitating samples as well.[4 8 10] However, many of these studies use samples that have limited statistical power, collapse sexual minority subgroups (eg, lesbians and bisexuals) and are restricted to older adults who are in cohabitating relationships. The present study overcomes those challenges with a large sample allowing for improved statistical power and stratified sexual minority subgroups during late adolescent and early adulthood.

Our findings support the previous research that has identified lower HRQL among sexual minorities[14–19] using other quality of life measures.[31–35] Bisexual and lesbian women in the USA-based NHS2 cohort had lower HRQL scores than heterosexuals.[15] Certain sexual minority groups, including bisexual women and heterosexual men with same-sex partners, had lower HRQL scores than their heterosexual peers without same-sex partners in the state-based Representative California Quality of Life Survey.[16] Men who have sex with men in a Swedish sample had lower psychosocial HRQL scores than their heterosexual peers.[17] Other studies among college students have also identified lower quality of life among sexual minority women and men, including one sample from Nigeria[18] and another from Cuba, Norway, India and South Africa.[19] Another study from a USA-based sample in Washington state was restricted to sexual minorities, and therefore lacked a heterosexual comparison group; in this sample, young bisexual women had worse HRQL than their lesbian peers, but these patterns reversed during mid-life when lesbians had worse HRQL.[36] Research on sexual orientation-related HRQL disparities using the EQ-5D-5L is scarce, but this measure was used in two recent publications, one from the GUTS cohort[14] and another based on a sample in Barcelona, Spain.[37] The current findings support these two publications' findings that sexual minorities participants presented worse HRQL than heterosexuals, and

the current study also adds new insights. For example, the previous GUTS publication did not include estimates of the different HRQL dimensions—nor the contribution from employment and healthcare—and the Spanish sample was not large enough to examine sexual minority subgroups.

In addition to the differences observed in the current study comparing sexual minorities with heterosexuals, there were also notable differences comparing men and women, as well as across sexual orientation subgroups. Male participants were more likely than female participants to have lacked a physical exam within the last year, but less likely than female participants to have experienced pain/discomfort (one of the five HRQL dimensions). These patterns align with existing literature on men being less likely than women to seek healthcare,[38] and the prevalence of pain that women experience compared with men.[39] While the primary analyses examined sexual minority subgroups in relation to the completely heterosexual group, some striking patterns emerged comparing sexual minority subgroups with one another. For example, bisexual women were the most likely to have been uninsured (13.8%) compared with completely heterosexual (3.9%) women, as well as compared with mostly heterosexual (5.7%) and lesbian (4.4%) women (p<0.001). These findings align with the literature documenting that bisexuals often experience some of the highest burdens of adverse health, even compared with other sexual minority subgroups.[40]

The GUTS cohort is made up of children of the NHS2 participants, so the results in this sample may not generalise to other populations; this cohort is predominantly of white race/ethnicity, their mothers are all nurses and the annual household income of the majority of the participants during their childhood was ≥$75000. Given this high social status, our estimates may underestimate the prevalence of unemployment, lack of health insurance, lack of healthcare access and poor HRQL. However, the fact that these disparities are pervasive in a cohort that predominantly holds high social status is striking. While our focus was on employment, health insurance and healthcare access, there may be other factors that mediate the sexual orientation-related HRQL disparities, including bullying victimisation, social status and others.[14] The lack of explanation in these disparities by HRQL in this cohort may be due to the participants' young age. Future research should explore how employment and healthcare may interact with age to drive worse HRQL as people age. Data were cross-sectional and limited on some of our variables such as health insurance and healthcare access. Future longitudinal studies could explore more detailed types of health insurance coverage as well as other measurements of healthcare access.

This study has a number of strengths, including the large sample drawn from young adults living across the USA. Building off the recently published data of sexual orientation-related disparities of HRQL index scores,[14] this is the first study to examine these disparities across

HRQL domains and sexual minority subgroups while also evaluating the role of additional factors including employment, health insurance and healthcare access. Using the EQ-5D-5L to measure HRQL allows the findings to be incorporated into cost-effectiveness research, which can inform public policy decisions.

These sexual orientation-related disparities in employment and health insurance in a population with high social status highlight the ubiquity of sexual orientation inequities in the employment and healthcare systems. The US Supreme Court's recent expansion of marriage rights to adults nationwide in same-sex relationships should lessen some of the sexual orientation-related disparities in health insurance. However, the adverse effects of previous bans are likely to persist.[41 42] Additionally, 28 states across the USA currently have no employment non-discrimination law covering sexual orientation—3 of these states have laws preventing the passage or enforcement of local non-discrimination laws.[20] Until all people, regardless of sexual orientation, are treated equally in the eyes of the law, including with non-discrimination laws protecting employment as well as housing, public accommodations and credit/lending, sexual orientation-related health disparities will persist.

**Author affiliations**
[1]Division of Adolescent/Young Adult Medicine, Boston Children's Hospital, Boston, Massachusetts, USA
[2]Department of Pediatrics, Harvard Medical School, Boston, Massachusetts, USA
[3]Channing Division of Network Medicine, Department of Medicine, Brigham and Women's Hospital, Boston, Massachusetts, USA
[4]Department of Epidemiology, Harvard TH Chan School of Public Health, Boston, Massachusetts, USA
[5]Division of General Pediatrics, Boston Children's Hospital, Boston, Massachusetts, USA
[6]The Fenway Institute, Fenway Health, Boston, Massachusetts, USA
[7]Department of Social and Behavioral Sciences, Harvard TH Chan School of Public Health, Boston, Massachusetts, USA

**Contributors** BMC conceptualised the project, supervised the analyses, and led the development and writing of the article. SBA supervised the data collection and, along with ARG, SLR and MS, aided in the interpretation of data and critically reviewed the manuscript for important intellectual content. VS conducted the analyses.

**Funding** BMC was supported by grant F32HD084000 and SBA by R01HD057368 and R01HD066963 from the Eunice Kennedy Shriver National Institute of Child Health and Human Development, National Institutes of Health. Additional funds were provided to BMC by GLMA: Health Professionals Advancing LGBT Equality's Lesbian Health Fund and by MRSG CPHPS 130006 from the American Cancer Society. SLR was partly supported by grant CER-1403-12625 from the Patient-Centered Outcomes Research Institute.

**Competing interests** None declared.

**Patient consent** Obtained.

**Ethics approval** This study was approved by the Brigham and Women's Hospital Institutional Review Board.

**Provenance and peer review** Not commissioned; externally peer reviewed.

**Data sharing statement** No additional data available.

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
