## [Reviewer comments · BMJ Open]

ARTICLE DETAILS

TITLE (PROVISIONAL)	Sexual orientation-related disparities in employment, health insurance, healthcare access, and health-related quality of life: a cohort study of U.S. male and female adolescents and young adults
AUTHORS	Charlton, Brittany; Gordon, Allegra; Reisner, Sari; Sarda, Vishnudas; Samnaliev, Mihail; Austin, Bryn

VERSION 1 – REVIEW

REVIEWER	Dr Edward McCann Trinity College Dublin Republic of Ireland
REVIEW RETURNED	16-Nov-2017

GENERAL COMMENTS	The study investigated sexual orientation-related disparities in a longitudinal cohort of US males and females. The abstract clearly sets out the study objectives and describes the setting, participants and the outcome measures used. The results and conclusions are succinctly articulated. The strengths and limitations of the study are considered. The study is timely in that it is the first to examine identified disparities, HRQL and evaluate the role of employment, health insurance and access to appropriate healthcare. The social exclusion, anti-discriminatory and legislative factors are contemplated. In the introduction, the authors refer to adolescents and young adults. Do these terms need to be defined? Should they appear in the title of the paper? The methods seem clear. However, I would advise that someone who is adept at scrutinizing the statistical analysis and the results contained in the tables, be approached to make comment. The discussion considers the study findings in relation to existing literature on the subject. Comparisons and contrasts are made with relevant studies in the field. Perhaps make explicit the usefulness of the findings and implications for health and social care service provision.
---

REVIEWER	Kirsty Clark UCLA School of Public Health, Los Angeles, USA
REVIEW RETURNED	03-Jan-2018

GENERAL COMMENTS	Major Revisions: 1) Overall, this is an interesting paper examining sexual orientation-related disparities in employment, health insurance, and healthcare access while evaluating their contributions to HRQL disparities. The analysis is strong, and there are potential policy implications of the
--

	findings. However, as it stands, the discussion doesn't fully reflect the implications of the results. For instance, there is no discussion of the apparent sex differences in HRQL dimensions or unemployment/uninsurance. As an example, in Table 1 we see that nearly 45% of lesbians experienced pain compared to just 24% of gay men. While the authors have clearly outlined the disparities between heterosexuals vs. sexual minorities, the interaction of sex and sexual orientation is overlooked in the discussion. A cogent discussion of these sex differences is warranted. Similarly, the authors do not include nuanced discussion of differences between sexual minority subgroups (e.g. lesbians vs. bisexuals), and potential implications (for example, in Table 2, bisexual women compared to lesbians are at greater risk of being unemployed/uninsured). The discussion requires additional detail to flush out the authors' unique findings, and contextualize implications of sex differences and sexual minority subgroup differences. Minor Revisions:  1) The title of the paper should remove "longitudinal cohort" as this analysis utilizes cross-sectional data 2) The introduction should include a sentence or two explaining how HRQL has been measured in prior studies, or if there is a standardized measure of HRQL and what that includes. The introduction mentions, "HRQL, including its multiple dimensions". It is currently unclear what those multiple dimensions are. 3) The sentence in the introduction beginning, "For example, over half of states..." should be restructured – it is currently unclear what "policy changes" that health insurance, HC access and HRQL can provide insight into. The authors seem to be referring to the lack of employment non-discrimination laws, but this is clunky phrasing. 4) The second paragraph in the discussion explains, "Women in same-sex couples benefit the least from employment as a way out of poverty because even among employed groups, women are at the highest risk of being in poverty [6]. The present study's findings support these data among sexual minority women." Please explain in greater detail how the present study supports these data.
--	---

REVIEWER	Chris Graham Picker Institute Europe, United Kingdom
REVIEW RETURNED	16-Feb-2018

GENERAL COMMENTS	This was a clear and well written study that used appropriate methods to address a straightforward set of objectives. I have around five substantive comments about study design and interpretation as well as a similar number of minor comments. These are listed below and I imagine that all should be quite easy to address.  1. Article summary, line 45 - "findings may underestimate the prevalence of unemployment" etc. They likely do underestimate these compared to the general population but that's not what the study sets out to do. The important thing isn't the overall prevalence of these characteristics but the differential prevalence between people of different sexual orientations - so I would suggest rephrasing this point in a way that addresses this. I think it is important to acknowledge the limitation that the relative scale of disparity between people with different sexual orientations may differ in more versus less affluent communities, in different ethnic groups, and in older populations - ie there may well be other interactions that
--

aren't measured here because of the comparatively homogeneous cohort. This is picked up in the discussion (p12, lines 7-14) but should also be articulated in the summary.

2. p3, lines 14-18 - "Until all people..." to "disparities will persist". This goes beyond the findings of the study to make a rhetorical argument for the importance of equalities legislation. It is a leap because the study does not show that either a) the observed differences are associated with public policy or legislation or b) new laws would help to reduce inequalities. It is right that the authors should set out the importance of the work for policy makers and this could be done appropriately with some rewording of this sentence - eg to describe that the results show evidence of inequalities in employment, health insurance, and HRQoL, and thus contribute to a case for promoting equality through new legislation to protect minority groups.

3. p4, introduction, lines 4-5 - "Half of all sexual minorities... report employment discrimination in their lifetime[1]". I don't think this is an accurate description of the report cited, which includes a range of estimates from definite studies but typically indicated that under half of LGB people report employment discrimination. For example, on p4 "42% of ... LGB-identified people had experienced at least one form of employment discrimination because of their sexual orientation at some point in their lives". The statistic that is over half is for the subset of LGB-identified people who are open about their sexual orientation in the workplace - for that group the figure from the 2008 General Social Survey is 56%. In the interests of balance, you might also note that the reported estimates of LGB & LGBT people's experiences of employment discrimination vary widely, and that many are based on very small and/or non-probability samples.

4. p9-10, tables 3 and 4 - these show that taking account of unemployment, health insurance status, and routine physical examination have a negligible effect on estimates of the association between sexual orientation and HRQoL. This seems important and somewhat unexpected, but is given very little attention in the discussion on p11-12. If there is something special about sexual orientation that leads to lower HRQoL regardless of employment, health insurance, and healthcare access, couldn't one argue that legislating to improve equalities in these areas would be unlikely to lead to improvements in LGB peoples' HRQoL? That's contrary to the suggestion in the discussion at the moment so needs to be considered. In fact, I think one might consider it less surprising given the cohort of younger adults - hypothetically it could be that lower access to employment and healthcare interacts with age to drive worse HRQoL as people get older. That would also seem broadly consistent with the findings reported in [36].

5. p12 - "No previous research, outside of the GUTS cohort[14], has used the EQ-5D-5L". This isn't quite true, but it's certainly fair to say that there is a dearth of published evidence about this. The authors should review Marti-Pastor et al (2018; doi: 10.1371/journal.pone.0191334), which uses EQ-5D-5L to look at differences in HRQoL by sexual orientation in Barcelona, Spain (note that this is a very recent publication that would not have been available at the time of submission). Also, England's GP Patient Survey - a massive and effectively general population study - also collects data on sexual orientation and EQ-5D-5L, but no peer reviewed studies have been published on this yet. However, Urwin &

	Whittaker (2016; doi: 10.1136/bmjopen-2016-011633) do use the survey to show that lesbian women are less likely to have recently seen a GP and gay men more likely compared to heterosexuals, which is consistent with the findings of this study. Similarly Elliott et al (2015; doi: 10.1007/s11606-014-2905-y) show associations between sexual orientation and self-reported health status in an earlier version of the survey without EQ-5D-5L. Other minor issues:  1. Abstract, line 9 - typo ("an ongoing a U.S.-based..." - "a" should be removed) 2. Abstract, line 10 - "and is primarily white and of middle-to-high socioeconomic positions" - phrasing could be improved, eg by saying "and is predominantly composed of people who are white and of middle-to-high socioeconomic positions" 3. Abstract, line 40 - "disparities... are pervasive... even in U.S. families of middle-to-high socioeconomic positions". Remove the word 'even', because this is the only population that the study addresses. It's not clear that results should be considered generalisable to a wider population. This also applies to p3, line 13, and p12 line 29. 4. p4, line 46, and p5, line 10 - it would be worth highlighting that the level of missing data for sexual orientation appears to be very low (~0.7%). Readers familiar with social surveys questions on sexual orientation may expect a much higher rate of nonresponse or refusal for this item, and as people whose sexual orientation is unknown are excluded it is a strength that this group is very small. 5. p6, Results - I would have found it helpful to have a summary of the characteristics of the whole cohort as a table rather than as narrative. This is particularly true for some of the values not presented, eg the overall proportion with less than full health - without this the onus is on the reader to do some of the interpretation.
--	---

VERSION 1 – AUTHOR RESPONSE

Review #1, Comment #1: The study investigated sexual orientation-related disparities in a longitudinal cohort of US males and females. The abstract clearly sets out the study objectives and describes the setting, participants and the outcome measures used. The results and conclusions are succinctly articulated. The strengths and limitations of the study are considered. The study is timely in that it is the first to examine identified disparities, HRQL and evaluate the role of employment, health insurance and access to appropriate healthcare. The social exclusion, anti-discriminatory and legislative factors are contemplated. The methods seem clear.

We appreciate the reviewer's positive remarks about the paper. After incorporating the revisions outlined below, we hope to have addressed the issues the editorial staff and reviewers highlighted.

Review #1, Comment #2: In the introduction, the authors refer to adolescents and young adults. Do these terms need to be defined? Should they appear in the title of the paper?

An age range for adolescent and young adults has now been added to the introduction and these words now appear in the paper's title, which reads "Sexual orientation-related disparities in employment, health insurance, healthcare access, and health-related quality of life: a cohort study of U.S. male and female adolescents and young adults."

Review #1, Comment #3: The discussion considers the study findings in relation to existing literature on the subject. Comparisons and contrasts are made with relevant studies in the field. Perhaps make explicit the usefulness of the findings and implications for health and social care service provision.

We have revised text in the introduction to make the usefulness of the findings more explicit. That text now reads: "Using research to document the downstream consequences of unemployment—health insurance, healthcare access, and HRQL—can aid policy makers in crafting the necessary legal changes to lessen these inequities, such as federal employment non-discrimination laws." Additionally, the discussion's concluding paragraph helps to contextualize the implications: "These sexual orientation-related disparities in employment and health insurance in a population with high social status highlight the ubiquity of sexual orientation inequities in the employment and healthcare systems. The U.S. Supreme Court's recent expansion of marriage rights to adults nationwide in same-sex relationships should lessen some of the sexual orientation-related disparities in health insurance. However, the adverse effects of previous bans are likely to persist [Hatzenbuehler et al. 2010 and 2012]. Additionally, 28 states across the U.S. currently have no employment non-discrimination law covering sexual orientation—3 of these states have laws preventing the passage or enforcement of local non-discrimination laws [Movement Advancement Project, accessed 2017]. Until all people, regardless of sexual orientation, are treated equally in the eyes of the law including with non-discrimination laws protecting employment as well as housing, public accommodations, and credit/lending, sexual orientation-related health disparities will persist."

Review #2, Comment #1: Overall, this is an interesting paper examining sexual orientation-related disparities in employment, health insurance, and healthcare access while evaluating their contributions to HRQL disparities. The analysis is strong, and there are potential policy implications of the findings. However, as it stands, the discussion doesn't fully reflect the implications of the results. For instance, there is no discussion of the apparent sex differences in HRQL dimensions or unemployment/uninsurance. As an example, in Table 1 we see that nearly 45% of lesbians experienced pain compared to just 24% of gay men. While the authors have clearly outlined the disparities between heterosexuals vs. sexual minorities, the interaction of sex and sexual orientation is overlooked in the discussion. A cogent discussion of these sex differences is warranted. Similarly, the authors do not include nuanced discussion of differences between sexual minority subgroups (e.g. lesbians vs. bisexuals), and potential implications (for example, in Table 2, bisexual women compared to lesbians are at greater risk of being unemployed/uninsured). The discussion requires additional detail to flush out the authors' unique findings, and contextualize implications of sex differences and sexual minority subgroup differences.

We appreciate the reviewer's complimentary remarks. The following additional paragraph discussing differences between men and women as well as across sexual orientation subgroups is now included in the discussion: "In addition to the differences observed in the current study comparing sexual minorities to heterosexuals, there were also notable differences comparing males and females as well as across sexual orientation subgroups. Male participants were more likely than females to have lacked a physical exam within the last year but less likely than females to have experienced pain/discomfort (one of the five HRQL dimensions). These patterns align with existing literature on men being less likely than women to seek healthcare [Galdas et al.] and the prevalence of pain that women experience compared to men [Fillingim et al.]. While the primary analyses examined sexual minority subgroups in relation to the completely heterosexual group, some striking patterns emerged comparing sexual minority subgroups to one another. For example, bisexual women were the most likely to have been uninsured (13.8%) compared to completely heterosexual (3.9%) women as well as compared to mostly heterosexual (5.7%) and lesbian (4.4%) women; p-value <0.001. These findings align with the literature documenting bisexuals often experience some of the highest burdens of adverse health, even compared to other sexual minority subgroups [Gorman et al.]."

Review #2, Comment #2: The title of the paper should remove "longitudinal cohort" as this analysis utilizes cross-sectional data

The title has now been revised to incorporate this suggestion as well as those from Reviewer #1 about adding the words "adolescent" and "young adult." It now reads "Sexual orientation-related disparities in employment, health insurance, healthcare access, and health-related quality of life: a cohort study of U.S. male and female adolescents and young adults."

Review #2, Comment #3: The introduction should include a sentence or two explaining how HRQL has been measured in prior studies, or if there is a standardized measure of HRQL and what that includes. The introduction mentions, "HRQL, including its multiple dimensions". It is currently unclear what those multiple dimensions are.

The introduction text has been revised to now read: "But, most of these data depend on limited measures of quality of life including measures like the Medical Outcomes Study 36-Item Short-Form Health Survey (SF-36) which does not explicitly assess HRQL, including its multiple dimensions (e.g., pain/discomfort, anxiety/depression). Additionally, few of these studies have examined potential mediators of the sexual orientation-related disparities in HRQL."

Review #2, Comment #4: The sentence in the introduction beginning, "For example, over half of states..." should be restructured – it is currently unclear what "policy changes" that health insurance, HC access and HRQL can provide insight into. The authors seem to be referring to the lack of employment non-discrimination laws, but this is clunky phrasing.

We appreciate the reviewer's suggestion to clarify this phrasing as this text frames the study's implications. New text now reads: "Using research to document the downstream consequences of unemployment—health insurance, healthcare access, and HRQL—can aid policy makers in crafting the necessary legal changes to lessen these inequities, such as federal employment non-discrimination laws."

Review #2, Comment #5: The second paragraph in the discussion explains, "Women in same-sex couples benefit the least from employment as a way out of poverty because even among employed groups, women are at the highest risk of being in poverty [6]. The present study's findings support these data among sexual minority women." Please explain in greater detail how the present study supports these data.

We agree with the reviewer that revisions could help to clarify how the current study's findings compare to all of the previous research on sexual orientation-related unemployment disparities (rather than just a comparison to citation #6 [Badgett et al. 2013]) so we have revised that latter sentence to read: "The present study's findings support these data revealing higher unemployment among sexual minority women compared to heterosexual women."

Review #3, Comment #1: This was a clear and well written study that used appropriate methods to address a straightforward set of objectives. I have around five substantive comments about study design and interpretation as well as a similar number of minor comments. These are listed below and I imagine that all should be quite easy to address.

We appreciate the reviewer's positive remarks about this being a "clear and well written" and we hope to have strengthened it through the various revisions outlined below.

Review #3, Comment #2: Article summary, line 45 - "findings may underestimate the prevalence of unemployment" etc. They likely do underestimate these compared to the general population but that's not what the study sets out to do. The important thing isn't the overall prevalence of these characteristics but the differential prevalence between people of different sexual orientations - so I would suggest rephrasing this point in a way that addresses this. I think it is important to acknowledge the limitation that the relative scale of disparity between people with different sexual orientations may differ in more versus less affluent communities, in different ethnic groups, and in older populations - ie there may well be other interactions that aren't measured here because of the comparatively homogeneous cohort. This is picked up in the discussion (p12, lines 7-14) but should also be articulated in the summary.

The article summary has been revised to include some of the nuance we outline in the discussion about the data's ability to address relative and absolute differences. The bulleted summary text now reads: "Cohort has predominantly high social status so while findings can examine sexual orientation-related differences, they may underestimate the prevalence of

unemployment, a lack of health insurance, a lack of healthcare access, and poor HRQL compared to other populations (e.g., low social status, elderly).”

Review #3, Comment #3: p3, lines 14-18 - "Until all people..." to "disparities will persist". This goes beyond the findings of the study to make a rhetorical argument for the importance of equalities legislation. It is a leap because the study does not show that either a) the observed differences are associated with public policy or legislation or b) new laws would help to reduce inequalities. It is right that the authors should set out the importance of the work for policy makers and this could be done appropriately with some rewording of this sentence - eg to describe that the results show evidence of inequalities in employment, health insurance, and HRQoL, and thus contribute to a case for promoting equality through new legislation to protect minority groups.

Given Reviewer #1’s request to make the implications of these findings more explicit, we have kept the sentiment of this text in the discussion but made minor revisions to that concluding sentence so that it’s clear we mean that new legislation is one part of lessening sexual-orientation related health disparities. It now reads: “Until all people, regardless of sexual orientation, are treated equally in the eyes of the law including with non-discrimination laws protecting employment as well as housing, public accommodations, and credit/lending, sexual orientation-related health disparities will persist.”

Review #3, Comment #4: p4, introduction, lines 4-5 - "Half of all sexual minorities... report employment discrimination in their lifetime[1]". I don't think this is an accurate description of the report cited, which includes a range of estimates from definite studies but typically indicated that under half of LGB people report employment discrimination. For example, on p4 "42% of ... LGB-identified people had experienced at least one form of employment discrimination because of their sexual orientation at some point in their lives". The statistic that is over half is for the subset of LGB-identified people who are open about their sexual orientation in the workplace - for that group the figure from the 2008 General Social Survey is 56%. In the interests of balance, you might also note that the reported estimates of LGB & LGBT people's experiences of employment discrimination vary widely, and that many are based on very small and/or non-probability samples.

We agree with the reviewer that estimates of employment discrimination do vary, including based on the population subset. However, since this opening sentence is simply meant to lay the groundwork for the introduction, we didn’t feel it necessary to critique this literature but instead have revised that sentence to now read “Nearly half of all sexual minorities...”

Review #3, Comment #5: p9-10, tables 3 and 4 - these show that taking account of unemployment, health insurance status, and routine physical examination have a negligible effect on estimates of the association between sexual orientation and HRQoL. This seems important and somewhat unexpected, but is given very little attention in the discussion on p11-12. If there is something special about sexual orientation that leads to lower HRQoL regardless of employment, health insurance, and healthcare access, couldn't one argue that legislating to improve equalities in these areas would be unlikely to lead to improvements in LGB peoples' HRQoL? That's contrary to the suggestion in the discussion at the moment so needs to be considered. In fact, I think one might consider it less surprising given the cohort of younger adults - hypothetically it could be that lower access to

employment and healthcare interacts with age to drive worse HRQoL as people get older. That would also seem broadly consistent with the findings reported in [36].

New text has been added to the discussion to these points. It now reads: “The lack of explanation in these disparities by HRQL in this cohort may be due to the participants’ young age. Future research should explore how employment and healthcare may interact with age to drive worse HRQL as people age.”

Review #3, Comment #6: p12 - "No previous research, outside of the GUTS cohort[14], has used the EQ-5D-5L". This isn't quite true, but it's certainly fair to say that there is a dearth of published evidence about this. The authors should review Marti-Pastor et al (2018; doi: 10.1371/journal.pone.0191334), which uses EQ-5D-5L to look at differences in HRQoL by sexual orientation in Barcelona, Spain (note that this is a very recent publication that would not have been available at the time of submission). Also, England's GP Patient Survey - a massive and effectively general population study - also collects data on sexual orientation and EQ-5D-5L, but no peer reviewed studies have been published on this yet. However, Urwin & Whittaker (2016; doi: 10.1136/bmjopen-2016-011633) do use the survey to show that lesbian women are less likely to have recently seen a GP and gay men more likely compared to heterosexuals, which is consistent with the findings of this study. Similarly Elliott et al (2015; doi: 10.1007/s11606-014-2905-y) show associations between sexual orientation and self-reported health status in an earlier version of the survey without EQ-5D-5L.

We thank the reviewer for referring us to the Marti-Pastor publication that was just released. We're also eager to see the sexual orientation-related EQ-5D-5L results from England's GP Patient Survey when those results are published as well, particularly in light of similar patterns to U.S. samples that the reviewer outlined in the Urwin and Whittaker as well as Ellicott et al. publications. We have made a number of edits in light of these publications including with the following text in the discussion: “Research on sexual orientation-related HRQL disparities using the EQ-5D-5L is scarce but this measure was used in two recent publications, one from the GUTS cohort [Austin et al.] and another based on a sample in Barcelona, Spain [Marti-Pastor et al]. The current findings support those two publications findings that sexual minorities participants presented worse HRQL than heterosexuals while the current study adds new insights. For example, the previous GUTS publication did not include estimates of the different HRQL dimensions—nor the contribution from employment and healthcare—and the Spanish sample was not large enough to examine sexual minority subgroups.”

Review #3, Comment #7: Abstract, line 9 - typo ("an ongoing a U.S.-based..." - "a" should be removed)

We appreciate the reviewer's careful reading and have revised that text to now read: “Growing Up Today Study, a U.S.-based longitudinal cohort begun in 1996.”

Review #3, Comment #8: Abstract, line 10 - "and is primarily white and of middle-to-high

socioeconomic positions" - phrasing could be improved, eg by saying "and is predominantly composed of people who are white and of middle-to-high socioeconomic positions"

This text has now been revised to read as the reviewer suggested: "predominantly composed of participants who are white and of middle-to-high socioeconomic positions."

Review #3, Comment #9: Abstract, line 40 - "disparities... are pervasive... even in U.S. families of middle-to-high socioeconomic positions". Remove the word 'even', because this is the only population that the study addresses. It's not clear that results should be considered generalisable to a wider population. This also applies to p3, line 13, and p12 line 29.

All mention of "even" has been removed throughout the abstract and manuscript.

Review #3, Comment #10: p4, line 46, and p5, line 10 - it would be worth highlighting that the level of missing data for sexual orientation appears to be very low (~0.7%). Readers familiar with social surveys questions on sexual orientation may expect a much higher rate of nonresponse or refusal for this item, and as people whose sexual orientation is unknown are excluded it is a strength that this group is very small.

A note has now been added to the methods section about the small amount of missing data on sexual orientation.

Review #3, Comment #11: p6, Results - I would have found it helpful to have a summary of the characteristics of the whole cohort as a table rather than as narrative. This is particularly true for some of the values not presented, eg the overall proportion with less than full health - without this the onus is on the reader to do some of the interpretation.

Given the paper's focus on sexual orientation-related disparities, we wanted to aid the reader in highlighting those results rather than on broader non-sexual orientation patterns. However, we now do refer the reader to a recent publication from our group that outlines these patterns without stratifying by sexual orientation (Austin et al. 2017).

VERSION 2 – REVIEW

REVIEWER	Dr Edward McCann Trinity College Dublin Republic of Ireland
REVIEW RETURNED	05-Mar-2018
GENERAL COMMENTS	Thank you for your revisions. I am satisfied that you have addressed all of the concerns.
REVIEWER	Kirsty Clark UCLA Fielding School of Public Health, Los Angeles, CA, USA

REVIEW RETURNED	06-Mar-2018
GENERAL COMMENTS	This is an interesting paper examining sexual orientation-related disparities in employment, health insurance, and healthcare access while evaluating their contributions to HRQL disparities. The analysis is strong, and there are potential policy implications of the findings. Nice work addressing reviewer comments.